# Thicker Retinal Nerve Fiber Layer with Age among Schoolchildren: The Hong Kong Children Eye Study

**DOI:** 10.3390/diagnostics12020500

**Published:** 2022-02-15

**Authors:** Xiu-Juan Zhang, Yi-Han Lau, Yu-Meng Wang, Hei-Nga Chan, Poemen P. Chan, Ka-Wai Kam, Patrick Ip, Wei Zhang, Alvin L. Young, Clement C. Tham, Chi-Pui Pang, Li-Jia Chen, Jason C. Yam

**Affiliations:** 1Department of Ophthalmology and Visual Sciences, The Chinese University of Hong Kong, Hong Kong SAR, China; zhangxiujuan@cuhk.edu.hk (X.-J.Z.); glorialau90@gmail.com (Y.-H.L.); ymwang917@gmail.com (Y.-M.W.); hnrubychan@cuhk.edu.hk (H.-N.C.); poemenchan@cuhk.edu.hk (P.P.C.); kamkawai.aziz@gmail.com (K.-W.K.); youngla@ha.org.hk (A.L.Y.); clemtham@cuhk.edu.hk (C.C.T.); cppang@cuhk.edu.hk (C.-P.P.); lijia_chen@cuhk.edu.hk (L.-J.C.); 2Division of Pediatric Ophthalmology and Strabismus, Tianjin Eye Hospital, Tianjin 300020, China; zhangwei3067@163.com; 3Hong Kong Eye Hospital, Kowloon, Hong Kong SAR, China; 4Department of Ophthalmology and Visual Sciences, Prince of Wales Hospital, Hong Kong SAR, China; 5Department of Paediatrics and Adolescent Medicine, LKS Faculty of Medicine, The University of Hong Kong, Hong Kong SAR, China; patrickiphk@gmail.com; 6Hong Kong Hub of Paediatric Excellence, The Chinese University of Hong Kong, Hong Kong SAR, China; 7Department of Ophthalmology, Hong Kong Children’s Hospital, Hong Kong SAR, China

**Keywords:** RNFL thickness, age correlation, children in Hong Kong, OCT measurement

## Abstract

This study aims to investigate the effect of age on the peripapillary retinal nerve fiber layer (p-RNFL) thickness among schoolchildren. A total of 4034 children aged 6–8 years old received comprehensive ophthalmological examinations. p-RNFL thickness was measured from a circular scan (⌀ = 3.4 mm) captured using spectral-domain optical coherence tomography (SD-OCT). Associations between p-RNFL thickness with ocular and systemic factors were determined by multivariate linear regression after adjusting potential confounders using generalized estimating equations (GEE). The mean global p-RNFL thickness was 106.60 ± 9.41 μm (range: 72 to 171 μm) in the right eyes, 105.99 ± 9.30 μm (range: 76 to 163 μm) in the left eyes, and 106.29 ± 9.36 μm (range: 72 to 171 μm) across both eyes. Age was positively correlated with p-RNFL after adjusting for axial length (AL) and confounding factors (β = 0.509; *p* = 0.001). Upon multivariable analysis, AL was positively associated with temporal p-RNFL thickness (β = 3.186, *p* < 0.001) but negatively with non-temporal p-RNFL thickness (β = (10.003, −2.294), *p* < 0.001). Sectoral p-RNFL was the thickest in the inferior temporal region (155.12 ± 19.42 μm, range 68 to 271 μm), followed by the superior temporal region (154.67 ± 19.99 μm, range 32 to 177 μm). To conclude, p-RNFL increased significantly with older age among children 6 to 8 years old in a converse trend compared to adults. Our results provide a reference for interpreting OCT information in children and suggest that stable p-RNFL thickness may not indicate a stable disease status in pediatric patients due to the age effects.

## 1. Introduction

Optic neuropathies in children lead to poor vision and even blindness as a result of conditions including glaucoma, optic nerve hypoplasia, and optic neuritis [1,2]. Early identification, classification, and monitoring of such diseases are crucial for early and effective treatment to preserve eyesight and prevent blindness. Structural investigation of the retinal layers by optical coherence tomography (OCT) is widely used. OCT is a non-contact medical imaging technology using reflected light to produce a detailed cross-sectional image of the eye [3,4]. It provides non-invasive, reproducible, high-resolution, and in vivo measurements of the retina and retinal nerve fiber layer (RNFL) for adults and children [5]. In particular, RNFL is an ocular structure containing ganglion cell axons, which are important components of the optic nerve. The attenuation of the peripapillary retinal nerve fiber layer (p-RNFL) is an early sign of loss of optic nerve tissue, which can be effectively detected by OCT.

While OCT investigations have been applied to children for the detection of eye diseases, a normative RNFL thickness database for children still needs to be established. As the eyeballs of children are growing, the patterns of change in RNFL with age are a useful reference for the detection of optic neuropathy in children. In adults, RNFL thinning in older age has been consistently reported [6,7,8,9]. On the contrary, the relation between RNFL thickness and age among children varies in different reports [10,11,12]. Several studies have shown that RNFL thickness does not correlate with age in children or young adults (age between 2 and 21 years) [12,13,14]. However, a recent study found that RNFL thickness increased with age in children aged <15 years old after adjusting for the ocular magnification effect [10]. The influence of age on RNFL thickness in children remains inconclusive. In addition, a significant difference in RNFL thickness among 6-year-old children of different ethnic groups has been reported [15]. We herein conducted a population-based study to evaluate p-RNFL thickness in children aged 6–8 years, in addition to its associations with age and other systemic and ocular factors.

## 2. Methods

### 2.1. Study Design and Population

The Hong Kong Children Eye Study (HKCES) is a population-based cross-sectional study targeting primary school children aged 6–8 years in Hong Kong, who were invited to visit the Chinese University of Hong Kong Eye Centre for comprehensive ophthalmological examinations and investigations. HKCES aims to evaluate various eye conditions among children in this age group and to determine the prevalence and risk factors of different eye diseases. Sample selection for the study was based on a stratified and clustered randomized sampling framework. We first stratified all primary schools registered with the Hong Kong Education Bureau according to the seven hospital clusters used by the Hospital Authority to organize services. After determining the target sample size, 5000 children were then invited to take part in this study, of which 4305 children participated. During the study period, 4273 children completed ophthalmological examinations, including cycloplegic refraction and visual acuity (VA) examinations. Sample size calculations and protocols for the HKCES have been previously reported [16]. The project adhered to the tenets of the Declaration of Helsinki, and approval was obtained from the Ethics Committee Board of the Chinese University of Hong Kong. Participating children and their parents gave informed consent prior to their participation in the study.

### 2.2. Ocular and Physical Examinations

Distance VA was measured using a logarithm of the minimum angle of resolution (logMAR) chart (Nidek Inc., Gamagori, Aichi, Japan). Best-corrected visual acuity (BCVA) was obtained through subjective refraction for all children with a logMAR score greater than 0.1 in either eye, performed by a trained optometrist using a trial frame placed and adjusted on the participants’ face. Refractive status was measured both before and after cycloplegia using an auto-refractor (Nidek ARK-510A, Gamagori, Japan) for each child. Two cycles of 1% cyclopentolate (Cyclogyl, Alcon-Convreur, Rijksweg, Belgium) and 1% tropicamide (Midorin-P, Santen, Osaka, Japan) were given 10 min apart. A third cycle of cyclopentolate and tropicamide drops was administered 30 min later if a pupillary light reflex was still present or the pupil size was less than 6.0 mm. An ophthalmologist would then examine in detail the anterior and posterior segments of each child using a slit-lamp (Haag-Streit, Koeniz, Switzerland) and binocular ophthalmoscope (Volk Optical Inc., Mentor, OH, USA). Spherical equivalent refraction (SER) was defined as spherical diopters (D) plus one-half the value for cylindrical diopters. Myopia was defined as SER ≤ −0.50 D, emmetropia as −0.50 D < SER < +0.50 D, and hyperopia as SER ≥ +0.50 D.

Ocular biometry, including axial length (AL) and corneal curvature measurements, was measured through non-contact partial-coherence laser interferometry (IOL Master; Carl Zeiss Meditec, Oberkochen, Germany). AL was measured as the distance from the anterior corneal vertex to the retinal pigment epithelium along fixation, automatically adjusted for retinal thickness. Central corneal thickness and intraocular pressure (IOP) were both measured using a Corvis ST tonometer (OCULUS Optikgeräte GmbH, Wetzlar, Germany). Blood pressure (BP) was measured using an automatic digital blood pressure monitor (Spacelabs Medical, Washington, DC, USA). Height and weight were measured using a professional integrated set (Seca, Hamburg, Germany). Body mass index (BMI) was calculated as body weight (in kilograms) divided by the square of body height (in metres).

### 2.3. OCT Imaging

OCT imaging was performed with a Spectralis SD-OCT (Heidelberg Engineering, Heidelberg, Germany) by a well-trained ophthalmic photographer. All the averaged B-scans in this study had a signal quality of at least 15 dB. p-RNFL imaging was performed at a central wavelength of 870 nm (Figure 1). The scan circle, with a diameter of approximately 3.45 mm (1536 A-scans), was manually positioned to locate the optic disc at the center of the circle, with the eye tracking system being activated. Fifteen B-scans were captured at the same location and were averaged automatically by the built-in software (Heidelberg Eye Explorer, version 1.6.1.0; Heidelberg Engineering, Heidelberg, Germany) to increase the image signal-to-noise ratio. The software also automatically segmented the p-RNFL for each averaged B-scan. The mean circumpapillary p-RNFL thickness and sectorial p-RNFL thickness were determined for six sectors: the nasal, temporal, superior nasal, inferior nasal, superior temporal, and inferior temporal regions.

### 2.4. Statistical Analysis

SPSS Statistics (version 24.0; IBM Corp., Armonk, NY, USA) was used for all statistical analyses. Confidence intervals (CIs) and *p* values (significant at a level of <0.05) were derived for the difference estimates, and regression models were applied with adjustments for cluster effects associated with the sampling design. In terms of descriptive statistics, numbers and percentages were reported for the categorical variables, whereas means and standard deviations (SDs) were reported for continuous variables, in addition to the 1st, 5th, 95th, and 99th percentiles. A Gaussian model was assumed for defining the 1st, 5th, and 95th percentile cutoff values for the mean global and sectoral p-RNFL thickness. To determine the associations of p-RNFL thickness with age, ocular, and systemic factors, linear regression models were applied using generalized estimating equations (GEE), with a view to account for inter-eye correlations within individuals in the multivariable analysis.

## 3. Results

### 3.1. Study Population

Among the 4273 children recruited, 159 children declined or could not complete the OCT examination. Another 80 children were excluded due to suboptimal imaging quality. Consequently, 8068 eyes from 4034 children were included in the final analysis, including 2067 boys (51.2%) and 1967 girls (48.8%), with a mean age of 7.61 ± 0.98 years (Table 1).

### 3.2. Normal Ranges of Global and Sectoral p-RNFL Thickness

The means for global p-RNFL thickness in the right and left eyes were 106.60 ± 9.41 μm (range: 72 to 171 μm) and 105.99 ± 9.30 μm (range: 76 to 163 μm), respectively. The mean was 106.29 ± 9.36 μm (range: 72 to 171 μm) for both eyes. Regarding the sectors in both eyes, the temporal inferior sector had the thickest mean p-RNFL (155.12 ± 19.42 μm, range 68 to 271 μm), while the nasal sector had the thinnest mean p-RNFL (66.11 ± 14.12 μm, range 32 to 177 μm; Table 2).

### 3.3. Associations of Global and Sectoral p-RNFL Thickness with Age

After adjusting for confounders, including age, gender, AL, BMI, systolic and diastolic BP, central corneal thickness, and IOP, an older age was associated with a thicker global p-RNFL (β = 0.509; 95% CI = (0.21, 0.80); *p* = 0.001; Table 3). This association with age was significant for p-RNFL thickness in the superior temporal (β = 0.953; 95% CI = (0.35, 1.56); *p* = 0.002), inferior temporal (β = 1.372; 95% CI = (0.76, 1.99); *p* < 0.001), superior nasal (β = 0.955; 95% CI = (0.29, 1.62); *p* = 0.005) and inferior nasal (β = 0.772; 95% CI = (0.06, 1.48); *p* = 0.03) regions (Table 4). Stratified by age, the mean global p-RNFL thickness was deceased by 6-, 7-, and 8-year age groups in myopia (*p* = 0.001), emmetropia (*p* = 0.028), and hyperopia (*p* = 0.023), respectively (Appendix A).

### 3.4. Associations of Global and Sectoral p-RNFL Thickness with Other Factors

Global p-RNFL thickness was negatively associated with AL (β = –2.917; 95% CI = (−3.23, −2.61); *p* < 0.001) and IOP (β = −0.115; 95% CI = (−0.18, −0.05); *p* < 0.001) (Table 3). BMI and female sex had a positive association with global p-RNFL thickness (β = 0.140; 95% CI = (0.03, 0.25); *p* = 0.013; β = 1.013; 95% CI = (0.42, 1.60), *p* = 0.001) (Table 3). AL was negatively associated with the sector p-RNFL thicknesses in all non-temporal quadrants (β = (−10.003, −2.294); *p* < 0.001), but positively associated with thickness in the temporal quadrant (β = 3.186; *p* < 0.001). Intraocular pressure was negatively associated with sectoral p-RNFL thicknesses for the temporal superior, temporal inferior, nasal superior, and nasal inferior regions (β = (−0.275, –0.192); *p* < 0.05) (Table 4).

## 4. Discussion

In this population-based study of healthy Chinese schoolchildren aged 6–8 years, we established a normative database of p-RNFL thickness. Notably, p-RNFL was thicker with age in children from 6 to 8 years old, a converse trend as that seen for adults [7]. Moreover, as AL increased, global p-RNFL thickness decreased, whereas temporal p-RNFL thickness increased. These findings indicate a different phenotype for p-RNFL in children as compared to adults. Furthermore, for pediatric patients with optic neuropathies, stable p-RNFL thickness may not indicate a stable disease status because of age in children.

### 4.1. Comparison of Age Correlation in Other Studies

Whereas the impact of age on RNFL thickness in adults has been widely regarded to be a negative correlation [17,18], its influence on RNFL thickness in children remains inconclusive. The studies by Yanni et al. [19] and Salchow et al. [20] reported similar decreases in RNFL thickness due to ageing among individuals in the age ranges of 5–15 years and 5–17 years, respectively. Nevertheless, most studies among different populations, including Mongolian, Hispanic, African-American, and Asian groups, found that RNFL thickness did not correlate with age in children or young adults aged between 2 to 21 years old (Table 5) [12,13,14,21]. A recent meta-analysis also concluded that the majority of reported studies showed no association between age and OCT parameters [22]. In the current study, we found that age was positively correlated with global p-RNFL thickness (*p* = 0.001). Our results are consistent with the positive correlation between ocular magnification-corrected average RNFL thickness and age in a study of 198 children under 15 years of age [10]. The variations between different studies may partly explained by different age rages, ethnics, and OCT measurements. The covariates that can be significantly associated with p-RNFL thickness should be adjusted. Hong and colleagues found that RNFL thickness measurements in children are likely to be overestimated if ocular magnification effects are not corrected [10]. In the current study, we adjusted known covariates such as Al, BMI, and IOP to ensure the accuracy of the results.

Among all sectors, we found that the superior and inferior sectoral p-RNFL parameters are, in particular, increased with age. These sectoral variations may be due to the larger numbers of nerve fibers converging to the optic nerve head from the superior and inferior arcuate bundles, relative to the numbers converging from the papillomacular bundle and nasal retina.

We propose that the increases in superior and inferior p-RNFL thickness among children could be attributed to an increase in axon diameter, glial cell proliferation, and/or formation of the radial peripapillary capillary network [10]. In pediatric patients, RNFL damage could be concealed by an increase in p-RNFL thickness with age. A stable p-RNFL thickness may therefore not necessarily indicate a stable disease status in pediatric patients with conditions such as pediatric glaucoma.

### 4.2. Comparison of Mean p-RNFL Thickness with Other Studies

The mean global p-RNFL thickness (106.29 ± 9.36 μm) for the schoolchildren in our study population was comparable to measurements obtained on healthy North American children aged 5–15 years (107.6 ± 1.2 μm) [19]. Similar results were also found among Asian children [10,12,23,24]. The mean global RNFL thickness was 107.71 ± 11.83 μm among Korean children aged 2–17 years [10] and 106.89 ± 12.84 μm among Shanghai Chinese students aged 6–11 years [23]. The published reports summarized in Table 5 indicate consistent mean global p-RNFL thickness in our and other studies.

### 4.3. Other Factors with Significant Correlations with p-RNFL

The significant correlation between p-RNFL thickness and AL in this study is consistent with previous reports [18,25]. The RNFL thinning related to a longer AL may be due to axial elongation in the posterior segment resulting from mechanical stretching by eyeball elongation [12]. In the current study, AL was correlated negatively with sectoral p-RNFL thickness in all non-temporal quadrants, but correlated positively with the temporal quadrant. This is consistent with previous studies among both adults and children [13,26]. There is likely a redistribution of retinal nerve fiber with increasing AL in myopia, in which the retinal thickness in the most central area—the temporal sector of the RNFL—is preserved, but the peripheral retina becomes thinner as it is less resistant to traction and stretch [23]. As the axial length increases, the retina is dragged towards the temporal horizon, and consequently the RNFL thickens in the temporal quadrant [27].

The topographic distribution pattern of RNFL thickness (temporal inferior > temporal superior > nasal inferior > nasal superior > temporal > nasal) found in this study has been consistently reported among normal children and adults [8,12]. The regional distribution of RNFL thickness corresponds with regional differences in the mean thickness of retinal ganglion cell axons in the retrobulbar part of the optic nerve, with thicker axons located in the inferior and superior regions, and the thinnest axons located in the temporal region [28]. The uneven sectoral distribution in RNFL thickness may be related to retinal anatomy, with the fovea being located about 0.5 mm inferior to a horizontal line drawn through the center of the optic disc [28]. This uneven distribution may lead to more retinal surface area and more retinal cells in the region inferior to the horizontal optic disc axis compared with the region superior to the axis.

Elevated IOP remains a major risk factor of glaucoma. IOP was significantly associated with rates of progressive RNFL loss in glaucoma patients [29]. However, the association of IOP with RNFL thickness in healthy adults and children is not clear. In this study, elevated IOP was related to thinner p-RNFL thickness in children. The true effect of IOP on p-RNFL thickness in a healthy population remains unknown. In a monkey model, there was a dynamic association between the high levels of IOP and RNFL thickness loss without drug intervention [30]. Future studies are needed to explore this relationship. Our results indicated that adjustments for IOP are necessary when making comparisons regarding p-RNFL thickness. Our study also found that global p-RNFL thickness in children did not decrease with higher blood pressure, but was positively related to a higher BMI. This finding is in contrast with studies on adults, in which the prevalence of localized RNFL defects was higher and the overall RNFL thickness was reduced among individuals with arterial hypertension and obesity [31,32,33]. One study showed a negative correlation between blood pressure (BP) and the mean global RNFL thickness, which suggested that good BP control among hypertensive subjects would help RNFL preservation [34]. A possible explanation for the discrepancy is that the damage of hypertension or obesity on arterioles of the retina and optic nerve head may be a long-term process. Lastly, girls have a greater myopic tendency and steeper corneal curvatures (i.e., shorter corneal radius of curvature) than boys [35]. The global RNFL thickness among females was 1 μm thicker than in males aged 40 to 80 years [36]. Estrogen production among females would lead to a protective effect on RNFL through estrogen receptors on RGCs [36]. This may partly explain the thicker p-RNFL observed for girls in our study population.

### 4.4. Strengths and Limitations

The strength of this study includes its large-scale population-based setting, in addition to its standardized randomized sampling method, which could eliminate sampling selection bias. Our final sample should be representative of the Chinese children population. Meanwhile, there are several limitations in the study. First, other optic disc parameters, such as disc size, have not been determined. Thus, the association between the size of the optic disc and p-RNFL thickness could not be explored. Second, the target age range of the children was 6–8 years. Our study population includes a narrow age band; hence, the study results may not be applicable to other populations. A large age range would provide more information on the trend of p-RNFL thickness with younger and older ages.

## 5. Conclusions

In conclusion, p-RNFL increases significantly with older age among children 6 to 8 years old. It is a converse trend compared to adults. A stable p-RNFL thickness may not warrant a stable disease status in pediatric patients. There is a different RNFL phenotype with age among children compared to adults. Our results provide a useful reference for interpreting tomography information in children.

## Figures and Tables

**Figure 1 diagnostics-12-00500-f001:**
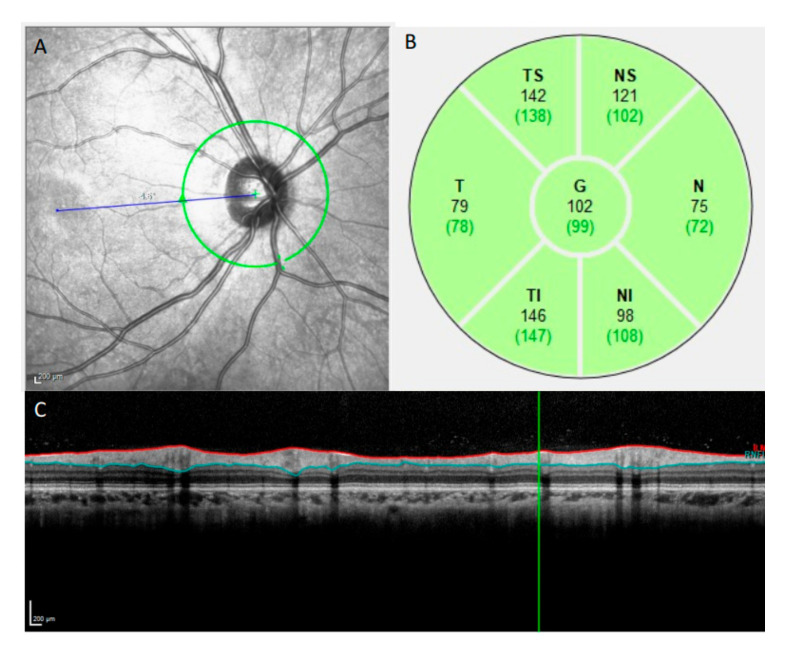
Report sample of the measurement of the retinal nerve fiber layer (RNFL) thickness with OCT. (**A**) A circle is drawn around the optic disc to measure the peripapillary RNFL thickness. (**B**) A picture demonstrating the RNFL. (**C**) Seven measurements were performed for each eye, providing the RNFL thickness of the nasal superior (NS), nasal (N), nasal-inferior (NI), temporal inferior (TI), temporal (T), temporal superior (TS), and global (G) sectors.

**Table 1 diagnostics-12-00500-t001:** Baseline characteristics of the school children in this study.

Sex	
Male, N (%)	2067 (51.2%)
Female, N (%)	1967 (48.8%)
Age (mean ± SD), years	7.61 ± 0.98
Body mass index (mean ± SD), kg/m^2^	16.14 ± 2.88
Systolic blood pressure (mean ± SD), mm Hg	101.51 ± 11.13
Diastolic blood pressure (mean ± SD), mm Hg	64.50 ± 9.03
Visual acuity, logMAR unit	
Right eyes (Mean ± SD)	0.01 ± 0.05
Left eyes (Mean ± SD)	0.02 ± 0.05
Axial length (mm)	
Right eyes (Mean ± SD)	23.14 ± 0.94
Left eyes (Mean ± SD)	23.13 ± 0.94
Spherical equivalent, D	
Right eyes (Mean ± SD)	0.13 ± 1.56
Left eyes (Mean ± SD)	0.17 ± 1.57
Central corneal thickness, μm	
Right eyes (Mean ± SD)	549.50 ± 32.51
Left eyes (Mean ± SD)	551.27 ± 31.83
Intraocular Pressure, mmHg	
Right eyes (Mean ± SD)	15.57 ± 2.53
Left eyes (Mean ± SD)	15.86 ± 2.77
logMAR, logarithm of the minimum angle of resolution	

**Table 2 diagnostics-12-00500-t002:** Ranges of peripapillary retinal nerve fiber layer thickness in global and different sectors.

p-RNFL Thickness, μm	Right Eyes						Left Eyes						Both Eyes					
			Percentiles						Percentiles						Percentiles			
	Mean (SD)	Range	1st	5th	95th	99th	Mean (SD)	Range	1st	5th	95th	99th	Mean (SD)	Range	1st	5th	95th	99th
Global	106.60 (9.41)	(72 to 171)	85.34	92.00	122.00	131.25	105.99 (9.30)	(76 to 163)	84.84	91.50	121.50	130.16	106.29 (9.36)	(72 to 171)	85.25	91.75	121.75	131.00
Temporal superior	154.22 (20.53)	(65 to 269)	103.00	122.00	188.00	207.00	155.12 (19.42)	(61 to 258)	111.00	124.00	187.00	206.65	154.67 (19.99)	(61 to 269)	106.00	123.00	187.55	207.00
Temporal	83.98 (13.13)	(51 to 214)	60.00	66.00	106.00	123.00	81.99 (12.40)	(41 to 198)	58.00	64.00	104.00	118.65	82.99 (12.81	(41 to 214)	59.00	65.00	105.00	121.00
Temporal inferior	155.65 (19.11)	(88 to 271)	114.00	127.00	188.00	204.65	154.60 (19.72)	(68 to 235)	106.35	124.00	187.00	201.00	155.12 (19.42)	(68 to 271)	111.00	125.00	188.00	203.31
Nasal inferior	117.26 (23.69)	(46 to 231)	68.00	80.75	156.00	181.65	115.83 (23.45)	(38 to 247)	67.00	80.00	156.00	175.65	116.51 (23.47)	(38 to 247)	67.00	80.00	156.00	178.00
Nasal	68.34 (14.15)	(32 to 199)	40.00	48.00	92.00	107.65	63.88 (13.73)	(32 to 177)	39.00	45.00	87.00	101.00	66.11 (14.12)	(32 to 177)	39.69	46.00	90.00	104.00
Nasal superior	121.14 (20.75)	(45 to 212)	76.00	89.00	156.00	177.00	130.46 (22.45)	(41 to 234)	80.35	96.00	168.00	190.00	125.77 (22.01)	(41 to 234)	78.00	92.00	163.00	187.00

p-RNFL: peripapillary retinal nerve fiber layer.

**Table 3 diagnostics-12-00500-t003:** Association of ocular and systemic factors with global peripapillary retinal nerve fiber layer thickness.

Parameters	β	95% CI		*p* Values
Age	0.509	0.21	0.80	0.001
Sex *	1.013	0.42	1.60	0.001
Body mass index, kg/m^2^	0.140	0.03	0.25	0.013
Systolic blood pressure, mm Hg	−0.011	−0.05	0.02	0.547
Diastolic blood pressure, mm Hg	−0.001	−0.04	0.04	0.946
Axial length, mm	−2.917	−3.23	−2.61	<0.001
Intraocular Pressure, mmHg	−0.115	−0.18	−0.06	<0.001
CCT, μm	0.000	−0.01	0.01	0.898

β was adjusted for all the ocular and systemic factors in the table. *: male = 1, female = 2.

**Table 4 diagnostics-12-00500-t004:** Association of ocular and systemic factors with sectional peripapillary retinal nerve fiber layer thickness.

	Temporal Superior				Temporal				Temporal Inferior			
Parameters	β	95% CI		*p* Values	β	95% CI		*p* Values	β	95% CI		*p* Values
Age	0.953	0.35	1.56	**0.002**	−0.210	−0.61	0.19	0.305	1.372	0.76	1.99	**0.000**
Sex *	1.568	0.36	2.77	0.011	−2.637	−3.42	−1.85	**0.000**	−0.982	−2.19	0.22	0.110
Body mass index, kg/m^2^	0.133	−0.08	0.35	0.223	0.093	−0.05	0.23	0.195	0.190	−0.02	0.40	0.081
Systolic blood pressure, mm Hg	−0.053	−0.12	0.02	0.134	−0.005	−0.05	0.04	0.822	−0.011	−0.08	0.06	0.765
Diastolic blood pressure, mm Hg	0.058	−0.02	0.14	0.163	0.009	−0.04	0.06	0.745	−0.013	−0.10	0.07	0.761
Axial length, mm	−2.294	−2.96	−1.63	**0.000**	3.186	2.75	3.62	**0.000**	−2.682	−3.36	−2.00	**0.000**
Intraocular Pressure, mmHg	−0.228	−0.40	-0.05	**0.011**	−0.069	−0.18	0.04	0.229	−0.208	−0.38	−0.03	**0.019**
CCT, μm	−0.004	−0.02	0.01	0.667	−0.002	−0.01	0.01	0.707	0.000	−0.02	0.02	0.970
Age	0.772	0.06	1.48	**0.034**	0.175	−0.25	0.60	0.425	0.955	0.29	1.62	**0.005**
Sex *	3.633	2.24	5.02	**0.000**	2.375	1.54	3.21	**0.000**	4.486	3.17	5.80	**0.000**
Body mass index, kg/m^2^	0.214	−0.03	0.46	0.083	0.137	−0.01	0.28	0.060	0.123	−0.11	0.36	0.309
Systolic blood pressure, mm Hg	−0.015	−0.09	0.06	0.704	0.016	−0.03	0.06	0.535	−0.021	−0.09	0.05	0.568
Diastolic blood pressure, mm Hg	−0.003	−0.09	0.09	0.957	−0.027	−0.08	0.03	0.331	−0.014	−0.10	0.07	0.746
Axial length, mm	−10.003	−10.77	−9.23	**0.000**	−4.071	−4.53	−3.61	**0.000**	−6.305	−7.04	−5.57	**0.000**
Intraocular Pressure, mmHg	−0.275	−0.47	−0.08	**0.005**	−0.192	−0.32	−0.07	**0.003**	0.024	−0.17	0.22	0.807
CCT, μm	0.000	−0.02	0.02	0.989	−0.004	−0.02	0.01	0.496	0.008	−0.01	0.03	0.391

β was adjusted for all the ocular and systemic factors in the table. *: male = 1, female = 2.

**Table 5 diagnostics-12-00500-t005:** Summary of studies on global peripapillary retinal nerve fiber layer thickness and correlation with age in children.

Study	Ethnicity	Age, Years (Mean ± SD; Range)	Type of OCT	Mean Global RNFL Thickness (μm)	Correlation with Age
Present study	Chinese (n = 4034)	7.61 ± 0.98; 6–8	SD-OCT	106.29 ± 9.36	Positively correlated
Wang CY, et al. (2018) [12]	Chinese and Mongolian (n = 1565)	11.9 ± 3.5; 6–21	SD-OCT	101.3 ± 9.2	No correlation
Hong SW, et al. (2017) [10]	Korean (n = 198)	8.61 ± 3.12; 2–18	Stratus OCT	107.71 ± 11.83	Positively correlated
Kang MT, et al. (2016) [13]	Chinese (n = 2893)	7.1 ± 0.4; 5.7–9.1	SD OCT	102.01 ± 8.02	Not reported
Chen L, et al. (2013) [23]	Chinese (n = 2324)	12.82 ± 3.11; 6–17	OCT-iVue100	106.89 ± 12.84	No correlation
Yanni SE, et al. (2013) [21]	non-Hispanic, African American, Hispanic, and Asian (n = 83)	9.14; 5–15	SD OCT	107.6 ± 1.2	Negatively correlated
Zhu BD, et al. (2013) [22]	Chinese (n = 2105)	12.34 ± 0.58; 10–16	SD-OCT	103.08 ± 9.01	No correlation
Tsai DC, et al. (2011)	Taiwanese (n = 470)	7–12	SD-OCT	109.4 ± 10.0	No correlation
Huynh SC, et al. (2006)	Caucasian, African American, Asian, and Hispanic (n = 1369)	6.71 ± 0.4	Stratus OCT	103.7 ± 11.4	No correlation
Salchow DJ, et al. (2005) [22]	Hispanic, African American, Caucasian (n = 92)	9.7 ± 2.7; 4–17	Stratus OCT	107.0 ± 11.1	Negatively correlated
Neelam, P, et al. (2014)	Indian (n = 120)	10.8 ± 3.24 years (range 5–17)	Stratus OCT	106.11 ± 9.5	No correlation
Leung, et al. (2010)	Chinese (n = 104)	9.75; (6.08–17.58)	Optical OCT	113	No correlation
Al-haddad Christiane	Middle Eastern (n = 108)	10.7 ± 3.1	Cirrus OCT	95.6 ± 8.7	No correlation

SD OCT: Spectral domain optical coherence tomography.

## Data Availability

The data presented in this study are available in Table 1, Table 2, Table 3, Table 4 and Table 5.

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
