# Peer review of "Thicker Retinal Nerve Fiber Layer with Age among Schoolchildren: The Hong Kong Children Eye Study"

_diagnostics, 2022, doi:10.3390/diagnostics12020500_

Round 1
Reviewer 1 Report
This study is a population-based study evaluating RNFLT in a large number of children. It is considered that the study design, statistical analysis, and discussion are well-examined reports. However, some revisions are considered necessary for this manuscript to be published.
- This study concludes that RNFLT increases with age in children aged 6-8 years, which may have been due to some statistical error due to the study of such a narrow age group. For example, can 6-year-olds, and 6-year-old and 11-month-olds have the same rating for RNFLT? Did the authors consider by age of the moon?
- In the Table 4, the authors have performed multiple regression analysis on the six objective variables, but the test results are slightly different for each measurement site. The authors should consider whether this is true that the test results differ from site to site, or whether there is a type I error due to multiple statistical tests.
- In the discussion section, The authors state that there were reports of different results on the relationship between RNFLT and age in children, but the authors should discuss a little more about the reasons.
Author Response
Thank you for the decision letter informing us that our paper would have a chance of further enhancement for publication. We appreciate the comments from the reviewers. We have carefully studied each comment and revised the manuscript accordingly. Enclosed please kindly find the revised manuscript and a point-by-point response letter to the reviewers.
Many thanks for your kind processing and consideration of our revised manuscript.
Author's Reply to the Review Report
This study is a population-based study evaluating RNFLT in a large number of children. It is considered that the study design, statistical analysis, and discussion are well-examined reports. However, some revisions are considered necessary for this manuscript to be published.
Author reply: Thank you for your comments. We have carefully studied each comment and revised the manuscript accordingly.
- This study concludes that RNFLT increases with age in children aged 6-8 years, which may have been due to some statistical error due to the study of such a narrow age group. For example, can 6-year-olds, and 6-year-old and 11-month-olds have the same rating for RNFLT? Did the authors consider by age of the moon?
Author reply: Thank you very much for your comments. We apologize for not describing clearly. The parameter of age was treated as continuous variable and the minimal unit was month when determining its associations with p-RNFL thickness. We have added the description in the “Statistical Analysis” and “Results”. On the other hand, we totally agree that the age range was very narrow in the study. We have included it as the study limitations in the “discussion”
- In the Table 4, the authors have performed multiple regression analysis on the six objective variables, but the test results are slightly different for each measurement site. The authors should consider whether this is true that the test results differ from site to site, or whether there is a type I error due to multiple statistical tests.
Author reply: Thank you very much for your careful review. We apologized that we marked the wrong Table Number. The association Global p-RNFL thickness with systemic and ocular parameters was showed in Table 3, not Table 4. We have corrected it. The results between the tables and description in the manuscript are consistent by double checking. To present the results clearly, we have reorganized the paragraph. We described the results of association of Global p-RNFL thickness first, as showed in Table 3; followed by the results of six sectoral p-RNFL thickness showed in Table 4. We only described the β and P values for the parameters with statistical significance (P<0.05).
- In the discussion section, the authors state that there were reports of different results on the relationship between RNFLT and age in children, but the authors should discuss a little more about the reasons.
Author reply: Thank you very much for your comments. More discussions were included accordingly.
Reviewer 2 Report
This is an article entitled “Thicker Retinal Nerve Fiber Layer with Age Among Schoolchildren: The Hong Kong Children Eye Study (diagnostics-1587898)” which evaluates the effect of age on the peripapillary retinal nerve fiber layer thickness among schoolchildren.
Please do no start the sentences with numbers. Revise throughout the manuscript.
Abstract
- Please add the ranges of all data.
Introduction
Methods
- Please give the generic name of tropicamide as well.
Results
- Please add the ranges of all data.
Discussion
Tables
- Please add the ranges of all data.
References
- good.
Author Response
Thank you for the decision letter informing us that our paper would have a chance of further enhancement for publication. We appreciate the comments from the reviewers. We have carefully studied each comment and revised the manuscript accordingly. Enclosed please kindly find the revised manuscript and a point-by-point response letter to the reviewers.
Many thanks for your kind processing and consideration of our revised manuscript.
Author's Reply to the Review Report
This is an article entitled “Thicker Retinal Nerve Fiber Layer with Age Among Schoolchildren: The Hong Kong Children Eye Study (diagnostics-1587898)” which evaluates the effect of age on the peripapillary retinal nerve fiber layer thickness among schoolchildren.
Please do no start the sentences with numbers. Revise throughout the manuscript.
Author reply: Thank you very much for your comments. We have revised accordingly.
Abstract
Please add the ranges of all data.
Author reply: Thank you for your comments. We have added the ranges of the data accordingly.
Methods
Please give the generic name of tropicamide as well.
Author reply: Thank you for your comments. We have added it accordingly.
Results
Please add the ranges of all data.
Author reply: Thank you for your comments. We have added the ranges of global and sectoral p-RNFL thickness in the results.
Tables
Please add the ranges of all data.
Author reply: Thank you for your comments. We have added the ranges of global and sectoral p-RNFL thickness in the tables.
References
good.
Author reply: Thank you for your favorable comments.